# Health-Related Quality of Life in Children: The Roles of Age, Gender and Interpersonal Trust

**DOI:** 10.3390/ijerph192215408

**Published:** 2022-11-21

**Authors:** Jing Wang, Wenjing Jin, Liping Shi, Yaoguo Geng, Xueli Zhu, Wanying Hu

**Affiliations:** 1School of Politics and Public Administration, Zhengzhou University, Zhengzhou 450001, China; 2School of Marxism, Zhengzhou University, Zhengzhou 450001, China; 3Department of Psychology, School of Social Development and Public Policy, Fudan University, Shanghai 200433, China; 4School of Education, Zhengzhou University, Zhengzhou 450001, China; 5Faculty of Psychology, Beijing Normal University, Beijing 100875, China

**Keywords:** health-related quality of life, children, age, gender, interpersonal trust

## Abstract

Health-related quality of life (HRQoL) is an interesting topic in health care sciences and psychology. Deeper insight into the internal mechanism of this effect through large samples is crucial to further understanding HRQoL and making targeted suggestions to improve HRQoL. The present study aims to investigate the mediating role of interpersonal trust between age and HRQoL from a developmental lens. The purpose of this study was to profile the Pediatric Quality of Life Inventory 4.0 generic scale in China and test the relationship between age and health-related quality of life, as well as the mediating role of interpersonal trust and the moderating role of gender. A sample of 6248 children completed measures of demography, health-related quality of life, and interpersonal trust. Regression analyses were performed to test the mediating role of interpersonal trust and the moderating role of gender. Age was associated with lower health-related quality of life and lower interpersonal trust. Similarly, gender differences were also noted, with boys reporting higher health-related quality of life and lower interpersonal trust than girls. Additionally, the health-related quality of life of girls declined more than that of boys with increasing age. Regression analyses revealed that age could predict decreased health-related quality of life via lower levels of interpersonal trust. What is more, the mediation effect was moderated by gender, with the observed mediation effect being stronger among boys than girls. The current study replicates age and gender differences in health-related quality of life and interpersonal trust. Moreover, this study explained how and when age affected the health-related quality of life of children, and provided a deeper understanding of the relation between age and health-related quality of life.

## 1. Introduction

The Program for a Healthy China 2030, approved by the Political Bureau of the CPC Central Committee, has set an action plan for the construction of a healthy China. The program aims to improve health, including health-related quality of life (HRQoL), because HRQoL is an indispensable prerequisite for national well-being as well as the foundation of economic and social development [1]. Health-related quality of life (HRQoL) is a multidimensional construct that can be defined as an individual’s perception of well-being in various life domains (including physical, psychological, and social components) in the context of their values, beliefs, expectations, goals, and cultural environment [2,3], reflecting an individual’s perceived physical and mental health over time [4]. Specifically, HRQoL describes an individual’s satisfaction or happiness in various life domains that affect or are affected by health [5]. In recent years, due to its influence on individuals and society, HRQoL has aroused the interest of the public health and psychology communities, and many studies on HRQoL have been conducted in youth populations. The study on HRQoL is especially suitable for children and adolescents due to the low mortality and morbidity in these groups, and they are in a rapid growth phase of development in which they may encounter various challenges [6]. In childhood and early adolescence, the tumultuous physical and social changes that accompany puberty, the desire for autonomy and distance from the family, and the transition from elementary school to middle school can all cause problems for children and adolescents. When adolescents are in settings (in school, at home, or in community programs) that are not attuned to their needs and characteristics, they can lose confidence in themselves and slip into negative healthy outcomes [7]. Health-related quality of life in children and adolescents is the basis for health-related quality of life and health in adulthood; therefore, knowing more about HRQoL in children and adolescents is of special relevance in public health [8]. Deeper insight into the internal mechanism of the development of HRQoL through large samples is crucial to further understanding HRQoL and making targeted suggestions to improve HRQoL. Population studies reported age differences in HRQoL in children and adolescents. For example, Bisegger and colleagues (2005) gathered data on 3710 youths from seven European countries and found that children have higher HRQoL than adolescents in many aspects [8]. Michel et al. (2009) analyzed normative data on more than 21,590 children and adolescents from 12 European countries and found that children generally showed better HRQoL than adolescents [3]. Recently, Meade and Dowswell (2015) examined data on 1111 adolescents from Australia and found that younger adolescents reported higher HRQoL than older adolescents [2]. Similar findings were reported in a North American context. Simon et al. (2008) investigated the determinants of HRQoL among a sample of children, and results revealed that HRQoL declines with age [9]. These may be related to environmental and physical factors. As they grow older, they are faced with changes in their physical and social lives and need to constantly adapt, and they experience more pressure. This instability will lead to a higher health-related burden [3].

Research assessing gender differences in HRQoL in children and adolescents is much more consistent across different studies. Specifically, a study on the self-reported problems of 9 to 17-year-old youth from seven European countries revealed that a greater decrease in HRQoL was found for girls than boys [10]. Another study on youth from 12 European countries revealed that girls often score lower in HRQoL than their male counterparts [3]. Similarly, Meade and Dowswell (2015) have noted that girls tend to have lower scores in HRQoL than boys [2]. Additionally, in an Iranian context, Jalali-Farahani (2016) and colleagues found that both self-reported and parent-reported scores in HRQoL in girls were significantly lower than in boys [10]. Gender differences may be related to the physiological characteristics of adolescents. Because girls are in the early stage of menstruation, fluctuations in hormones, the frequent occurrence of stressful life events, and specific coping mechanisms may lead to the deterioration of mental health, and girls are more concerned about their well-being than boys [11]. In addition, girls and boys deal with problems in different ways [10]. Therefore, the gender difference in HRQoL may be related to their physiological mechanisms and coping styles.

Consistent with other international studies, a few studies have replicated both age and gender differences in HRQoL in a Chinese context. For example, Dou (2007) gathered data on 1113 children from Henan province and found that children in the elementary grades reported higher HRQoL than older children, and boys reported higher HRQoL than girls [12]. Li and colleagues (2015) gathered data on 860 children from Zhejiang province and found that elementary school students and boys reported higher HRQoL than their comparable counterparts [13]. Similarly, Jiang (2014) gathered data on 781 adolescents from Zhejiang province and found that boys reported higher HRQoL than girls [14]. These results suggest that age and gender differences were found across different studies and countries. However, these findings need to be corroborated with a larger sample size. Despite the recent surge in research examining the age and gender differences in HRQoL, several theoretical questions remain unanswered. First, it is unclear, from a developmental lens, how age and developmental predictors are associated with HRQoL, such as physical development, environmental factors, and interpersonal relationships. Second, potential developmental mediators and moderators have yet to be tested in relation to age and HRQoL. HRQoL was thought to be more influenced by objective factors, such as health conditions, socioeconomic position, and environment [15,16,17,18], a few studies have noted that subjective factors, such as interpersonal trust, were associated with HRQoL [19,20], but the subjects were limited to adults.

Interpersonal trust refers to a mental state of individuals involving confident, positive expectations about the actions of another [21] and is a positive predictor of health and well-being. Many studies found that higher levels of trust are associated with better health, better well-being, and lower mortality [20]. Conversely, lower levels of trust were found to be associated with psychological distress, loneliness, depression, and suicidal ideation [22,23,24,25] and associated with major causes of death, including heart disease, cancers, and violent death [26]. So, interpersonal trust is theorized as a core factor of social capital and a protective factor for health [27]. Theoretically, the Health Promotion Model (HPM) posits that individual characteristics and interpersonal trust are all influential factors of HRQoL, and individual characteristics are remote antecedents of HRQoL [28,29]. The Health Promotion Model focuses on people’s interactions with their physical and interpersonal environments during attempts to improve health [30]. Specifically, the model emphasizes the active role that a person has in initiating and maintaining health-promoting behaviors, and in shaping their own environment to support health-promoting behaviors [31].

Surprisingly, only two studies, to our knowledge, focus on the links between interpersonal trust and HRQoL. Specifically, Tokuda and colleagues (2008) examined the relationship between interpersonal trust and HRQoL among a sample of adults from a Japanese population [20]. Results revealed that individuals high in interpersonal trust were more likely to report that they had higher HRQoL than individuals low in interpersonal trust [20]. Liang (2015) investigated the aforementioned link in a sample of Chinese rural-to-urban migrants and found that interpersonal trust emerged as predictors for higher HRQoL [19]. The work of Tokuda et al. (2008) and Liang (2015) is influential in establishing associations between interpersonal trust and HRQoL [19,20]. However, because research on interpersonal trust and HRQoL is still in an early stage, it is necessary to record the possible relationships.

A growing body of research has indicated that interpersonal trust is pertinent to age and gender. Specifically, adults tend to have an increase in interpersonal trust as they get older [32,33], whereas children and adolescents tend to have a decline in interpersonal trust as they get older [34,35,36,37]. These results suggest that interpersonal trust might be an important developmental predictor, especially for youth. One of the possible reasons for this change may be that as youth attempt to navigate their lives, negative attitudes, including mistrust, are likely to develop. In the broader literature, there are gender differences in interpersonal trust. The results of trust research showed that women are more credulous than men [38], perhaps due to their greater striving for social affiliation [39].

For these reasons, it seems more likely that interpersonal trust is a mediator between age and HRQoL. However, no published work to date has merged these variables together to examine whether interpersonal trust mediates the association between age and HRQoL. Further, given the gender differences in HRQoL [2,9,10], it is necessary to investigate whether gender differences in HRQoL could be mediated by interpersonal trust.

### The Present Study

Although some studies have explored the relationship between age and HRQoL, there has been a lack of research exploring the potential mechanisms underlying the age-HRQoL pathway, especially among children [40]. In addition, there are gender differences in HRQoL and interpersonal trust. The study found that the gender differences in HRQoL between adolescents and college students are inconsistent. Specifically, among adolescents, boys reported better HRQoL than girls, but among college students, boys experienced worse HRQoL [41]. The current study aims to investigate the role of interpersonal trust as a mediator between age and HRQoL from a developmental lens. Based on previous reviews, we predicted that (1) younger participants and boys tend to have higher scores on HRQoL than older participants and girls; (2) age will predict lower HRQoL via interpersonal mistrust; and (3) gender will moderate the relations between age, interpersonal trust, and HRQoL. Compared with girls, the influence of age on interpersonal trust is more significant in boys; the relationship between age and HRQoL of girls is stronger than that of boys. Our hypothesized model can be seen in Figure 1.

## 2. Method

### 2.1. Participants

Participants were recruited from primary and secondary schools in China. Seven thousand students took part in this study in exchange for extra course credit in their psychology class. Of the initial 7000 students, 752 (11%) were excluded because of incomplete data related to study variables. The final sample consisted of 6248 Chinese students from 9 to 15 years of age (M = 12.96, SD = 1.72) enrolled in fourth grade to ninth grade, including 2938 girls (47%; M = 12.93, SD = 1.71) and 3310 boys (53%; M = 12.99, SD = 1.73). The distribution of age and gender among participants is presented in Table 1. There was no significant difference in age between the boys and girls (t = 1.57, *p* = 0.12), and no significant difference in the demographics between the students who were included in the study and those who were excluded.

### 2.2. Procedures

With appropriate permissions from the school boards, we fully communicated with parents/legal guardians through the teacher to obtain their assent. After obtaining the assent of teachers, parents, and students, students completed the survey during normal class hours. Students were allowed to participate voluntarily and anonymously and clarify questions in a unified environment. A cluster sampling method was employed to sample. Moreover, they were thanked and assured that their answers were completely confidential. All procedures performed in studies involving human participants were in accordance with the ethical standards of the institutional review board at Zhengzhou University and with the 1964 Helsinki Declaration and its later amendments or comparable ethical standards.

### 2.3. Measures

#### 2.3.1. Health-Related Quality of Life (HRQoL)

HRQoL was measured using the Pediatric Quality of Life Inventory 4.0 generic scale (PedsQL), which is a well-validated measure to evaluate the HRQoL of children. The scale comprises 23 items that are scored on a 5-point scale (100 points for never; 75 points for hardly ever; 50 points for sometimes; 25 points for often; 0 points for always) and provides the physical, emotional, social, and school functioning score. The score of each subscale was the sum of the scores of the contained items divided by the number of the contained items, and the score of the entire scale was the sum of the scores of the items divided by the number of the items in the whole scale. Higher scores indicate that the children have a better HRQoL. In this study, we used the Chinese version of this scale. Sample items included low energy and worry about what will happen, and the Cronbach’s α was 0.90 for the entire scale.

#### 2.3.2. Interpersonal Trust

Interpersonal trust was measured using the Kiddie Machiavellianism Scale (KMS), which is a validated measure to evaluate the children’s attitudes towards human nature and interpersonal trust in interpersonal relationships [42]. The scale comprises 16 items that are scored on a 4-point scale and provides a total score, with higher scores indicative of interpersonal mistrust. The Chinese version of the Kiddie Machiavellianism Scale was documented elsewhere. Sample items included anyone who completely trusts anyone else is asking for trouble, and you should always be honest, no matter what. In this study, the Cronbach’s α was 0.74 for the entire scale.

### 2.4. Data Analyses

All statistical analyses were carried out using SPSS Statistics for Windows, Version 22.0 (IBM Corp, Armonk, NY, USA). Boys and girls were compared on all constructs. Relationships among age, gender, interpersonal trust, and HRQoL were examined with Pearson’s correlation. Tests for the moderated mediation effect (Model 59) were conducted using the procedures outlined in Hayes [43]. Before regression analyses, all variables were standardized. In the regression model, age was entered as an independent variable, interpersonal trust as a mediating variable, gender as a moderating variable, and HRQoL as the outcome variable. We generated 5000 bias-corrected bootstrapped samples to estimate the confidence interval when doing moderated mediation effect analysis [44], with the 95% confidence interval without zero meaning statistical significance.

## 3. Results

### 3.1. The Score of HRQoL and Interpersonal Trust following Child Development

Significant trends were observed in both HRQoL and interpersonal trust related to age, indicating that, compared with older children, younger children have higher interpersonal trust and a higher quality of life. Specifically (Figure 2 and Figure 3), the total score of HRQoL in both sexes was increasing gradually from 9 years old to 11 years old and then decreasing sharply between 11 years old and 15 years old, with a peak score at 11 years old in our sample. As for interpersonal trust, the total score in boys was increasing gradually from 9 years old to 15 years old (which means that boys tended to have lower interpersonal trust as they grew older), whereas the total score in girls was decreasing gradually from 9 years old to 11 years old, and then increasing sharply between 11 and 12 years old, with a peak score at 15 years old in our sample (which means that girls tended to have lower interpersonal trust on the whole as they grew older).

### 3.2. Gender Differences

Table 2 shows the means and standard deviations of study variables. Compared to girls, boys had lower interpersonal trust (higher scores of interpersonal mistrust) and higher overall HRQoL.

### 3.3. Correlations

Pearson product-moment correlations are presented in Table 3. Age was associated with lower interpersonal trust (higher scores of interpersonal mistrust) and lower HRQoL. Gender (coded 1 = boys; 2 = girls) was associated with higher interpersonal trust (lower scores of interpersonal mistrust) and lower HRQoL, although the correlation is small. Gender (coded 1 = boys, 2 = girls) was negatively correlated with HRQoL, indicating that girls have a lower level of HRQoL. Gender was negatively correlated with physical function and emotional function and positively correlated with social function, reflecting differences in their perceptions of their functioning in these domains. Additionally, interpersonal mistrust was associated with lower overall HRQoL and all components of HRQoL.

### 3.4. Moderated Mediation Effect Analyses

We conducted a moderated mediating effect analysis to confirm whether the gender of the participant moderates the mediating effect of interpersonal trust on the relationships between age and HRQoL (Figure 4).

For the indirect effect of interpersonal trust on the relationship between age and overall HRQoL (Table 4), results revealed that age was directly associated with lower HRQoL (β = −0.17, *p* < 0.01, 95% CI [−0.20, −0.15]), and indirectly associated with lower HRQoL via lower interpersonal trust (higher scores of interpersonal mistrust) (95% CI [−0.08, −0.06]). These results suggest a significant mediating effect of interpersonal trust on this relationship.

For the moderating effect of gender on the relationship between age and HRQoL (Table 4), results revealed that age was associated with lower HRQoL (β = −0.17, *p* < 0.01, 95% CI [−0.20, −0.15]), gender was associated with lower HRQoL (β = −0.07, *p* < 0.05, 95% CI [0.05, 0.09]). The interaction of age and gender was negatively associated with HRQoL (β = −0.05, *p* < 0.05, 95% CI [−0.07, −0.02]). Results of the simple slope test revealed that the regression slope of age on HRQoL was stronger for girls (β = −0.28, *p* < 0.01) than boys (β = −0.21, *p* < 0.01) (Figure 5).

For the moderating effect of gender on the relationship between age and interpersonal trust (Table 4), the results revealed that age was associated with lower interpersonal trust (higher scores of interpersonal mistrust) (β = 0.26, *p* < 0.01, 95% CI [0.23, 0.28]), and gender was associated with higher interpersonal trust (lower scores of interpersonal mistrust) (β = −0.08, *p* < 0.05, 95% CI [−0.10, −0.05]). The interaction of age and gender was positively associated with interpersonal trust (β = −0.03, *p* < 0.05, 95% CI [−0.06, −0.01]). Results of the simple slope test revealed that the regression slope of age on interpersonal trust was stronger for boys (β = 0.28, *p* < 0.01) than girls (β = 0.22, *p* < 0.01) (Figure 6).

For the moderating effect of gender on the relationship between interpersonal trust and HRQoL (Table 4), the results revealed that interpersonal trust was associated with higher HRQoL (β = −0.29, *p* < 0.01, 95% CI [−0.31, −0.26]), and gender was associated with higher interpersonal trust (lower scores of interpersonal mistrust) (β = −0.08, *p* < 0.05, 95% CI [−0.10, −0.05]). The interaction of gender and interpersonal trust was not significantly associated with HRQoL (β = 0.02, *p* > 0.05, 95% CI [−0.04, 0.06]).

The conditional indirect effect was stronger for boys (a × b = −0.77, SE = 0.01, 95% CI [−0.09, −0.07]) than girls (a × b = −0.68, SE = 0.01, 95% CI [−0.08, −0.06]). This result meant that interpersonal trust played a mediating role in the relationship between age and HRQoL, and the mediating effect was moderated by the gender of the participants (Figure 4).

## 4. Discussion

HRQoL is an informative topic in health care sciences and psychology. Researchers have examined various objective and subjective correlates. Although age, gender, and health conditions are associated with HRQoL, there is a paucity of research examining developmental predictors [45,46,47]. The present study adds to the literature by examining the underlying psychological processes associated with HRQoL in children.

In good agreement with previous research, this study found a change trend in the development of HRQoL at different ages. Results revealed that age was negatively associated with HRQoL, suggesting that younger participants scored higher on HRQoL compared to older participants. This is different from the relationship between adult age and interpersonal trust. Middle-aged people’s social relationships (spouse, coworkers, and neighbors) seem to last longer than those formed when they are just adults. Similarly, many adults may be more deeply integrated into social groups where most members are connected [48]. At the same time, the current study also replicates gender differences in HRQoL [2,15]. Results revealed that boys reported higher scores on overall HRQoL than girls. Additionally, results of our study further revealed that the relations between age and HRQoL differed significantly between the sexes, with the negative correlation being stronger for girls than for boys, thus demonstrating that HRQoL of girls declined more than that of boys with increasing age. In a Chinese context, these findings are comparable to Australia, Iran, North America, and European norms [9,10] and contribute to the international data on similarities and differences in the Pediatric Quality of Life Inventory 4.0 generic scale (PedsQL). In adults, there are still significant differences in the quality of life between men and women, which are more closely related to objective factors such as marital status and family income [49]. The biological changes are associated with the transition into early adolescence. When the hormones are activated in early puberty, most children undergo a growth spurt, develop primary and secondary sex characteristics, become fertile, and experience increased sexual libido [7]. Girls begin to experience these pubertal changes earlier than boys, so girls and boys at the same chronological age are likely have different experiences, such as early maturing white girls have lower self-esteem and more difficulty adjusting to the transition from elementary to junior high school than boys [50]. The results of this study are consistent with previous studies. Interpersonal trust is a subjective factor closely related to HRQoL [19,20]. As expected, interpersonal trust was associated with higher overall HRQoL and all components of HRQoL. There was a positive association between interpersonal trust and HRQoL.

To our knowledge, the present study is one of the first to empirically test the mediating role of interpersonal trust on the association between age and HRQoL. Results revealed that age was directly associated with lower HRQoL, and indirectly associated with lower HRQoL via lower interpersonal trust. These results suggest that (1) with an increase in age, more participants endorsed lower interpersonal trust. This result is in line with previous research [36], which suggests that children become more suspicious of others as they get older. One possibility is that adolescence and young adulthood are associated with risk-taking and impulsivity, so negative attitudes such as suspiciousness, mistrust, negativity, and cynicism might be most apparent at this age [34,35]; (2) as proposed by Tokuda et al. (2008) and Liang (2015), the lower their interpersonal trust, the more participants endorsed lower HRQoL [19,20]; (3) older adolescents reported greater distrust of others and lower levels of HRQoL. That is, interpersonal trust mediated the effects of age on HRQoL. These results provide insight into the behaviors and underlying psychological processes associated with HRQoL and are theoretically important for our understanding of why age is associated with HRQoL.

As expected, the present study also found that the observed mediation model was moderated by the gender of the participant. These findings support the health promotion model that people’s health depends on the interactions of their physical and interpersonal environments [30], indicating that people’ gender interacting with their age not only predicted interpersonal trust but also predicted HRQoL. Specifically, the indirect effect of interpersonal trust was stronger for boys than for girls. Possible reasons for this gender difference include: (1) girls are more likely than boys to develop mutualistic social schemata, which include traits such as increased capacity for self-regulation and interpersonal sensitivity, and show more empathy and prosocial behaviors [19,51]. Conversely, boys are more likely than girls to develop antagonistic social schemata, which include traits such as limited empathy and a decreased interpersonal trust tendency, and show more impulsive and aggressive behaviors [52,53]. Although this antagonistic social schemata may be an adaption to solve life problems in the face of a harsh and unpredictable world [54], this tendency in turn could reduce HRQoL of boys [55]; (2) previous research has noted that compared to males, HRQoL of females was primarily influenced by objective factors, such as diverse social expectations, puberty (i.e., menstruation and fluctuating hormones), more frequent somatic symptoms [2], and stressful life events [56]. Briefly, the above moderated mediation effects mean that the relations between age and interpersonal trust are more substantial for boys than for girls and indicate that it is necessary to consider gender when investigating the relations between age, interpersonal trust, and HRQoL.

In conclusion, this study examines age and interpersonal trust in relation to HRQoL using the data of 6248 children. We also examined the potential moderating role of gender in the relationship. Consistent with our hypotheses, the results revealed that age was associated with lower HRQoL, interpersonal trust was associated with higher HRQoL, and age was indirectly associated with lower HRQoL via lower interpersonal trust. Notably, gender did serve as a moderator, with the indirect effect being stronger for boys than for girls. Our findings explained how and when age affected children’s HRQoL and extended the current literature on the predictors of age and gender differences in HRQoL of children.

Our paper has three implications for prevention and intervention for these youth. First, given that age was associated with lower HRQoL, youth can be taught useful skills (e.g., skills in communication and emotion regulation) in order to achieve a satisfactory life. Second, it is important to note that the HRQoL of girls declined more than that of boys as they grew older, indicating that researchers and practitioners should pay attention to how to improve the quality of life for girls. Third, because interpersonal trust was found to be associated with higher HRQoL, we speculate that having a generally high level of interpersonal trust will bring a more adaptive and productive life to individuals, especially in the Chinese context [26]. Here are some recommendations to improve interpersonal trust: Various organizations should be widely involved in promoting children’s interpersonal trust. Firstly, schools should give more psychological intervention to children and adolescents with low interpersonal trust, such as psychological outward-bound training and trust journeys. It can effectively improve interpersonal communication problems and enhance interpersonal trust; secondly, psychological consultants should pay more attention to gender differences and provide targeted suggestions suitable for the development of girls and boys; thirdly, families should timely explain the physiological knowledge of adolescence to teenagers, establish trust and close interpersonal relationships with teenagers, better help them solve problems, and improve their HRQoL.

## 5. Limitations

There are several limitations to the current study that should be considered. First, only self-reported measures were used. Future research may use multiple-source data, such as that from teachers and family members, to triangulate on the results found using self-reports. Second, the present study is of a cross-sectional design and reveals the correlativity among all variables [57,58]. Hence, experimental or longitudinal studies are needed to reveal the causality in the future. Third, the present study only examined interpersonal trust as the mediator in the relation between age and HRQoL. So, it is necessary to explore multiple mediators, such as resilience and social support, as well as other moderators in future studies to reveal how and when age affects HRQoL in children. Fourth, HRQoL is multifaceted. So, another limitation is that we only use the overall HRQoL in this study. Future research may investigate the associations between age, gender, interpersonal trust, and different types of HRQoL. Last but not least, recruitment was reliant on Chinese-speaking populations. The culture from which participants are recruited may influence interpersonal trust and willingness to provide socially desirable responses [59]. Under a Chinese cultural background, interpersonal factors will affect interpersonal trust to a large extent; that is, the cognition of intimate and sparse relationships will affect people’s interpersonal trust. In the context of western culture, the influence of individual constitution, such as ability and responsibility, is prior to interpersonal factors or independent of relationship factors. Future research should consider a more diverse population, such as Western populations.

## 6. Conclusions

The present study adds to the growing literature on the association between age and health-related quality of life. The results reveal that the older have low interpersonal trust and HRQoL, and the gender difference is significant. Therefore, researchers should not only pay attention to the objective factors affecting HRQoL but also pay attention to the important influence of interpersonal trust.

## Figures and Tables

**Figure 1 ijerph-19-15408-f001:**
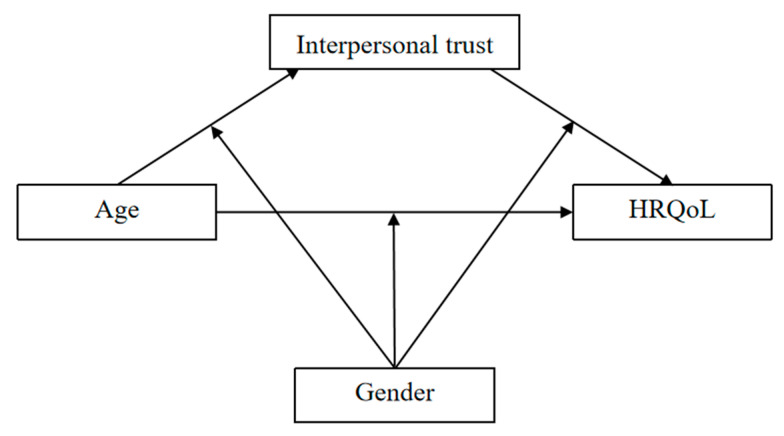
Hypothesized model.

**Figure 2 ijerph-19-15408-f002:**
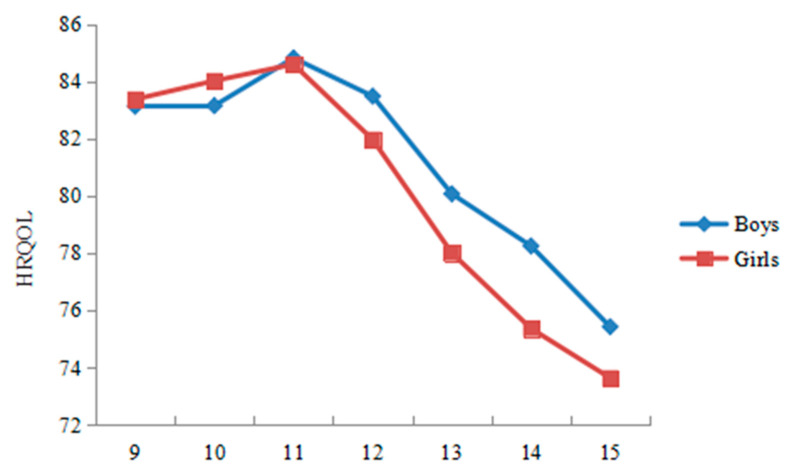
The total score of health-related quality of life by age and gender.

**Figure 3 ijerph-19-15408-f003:**
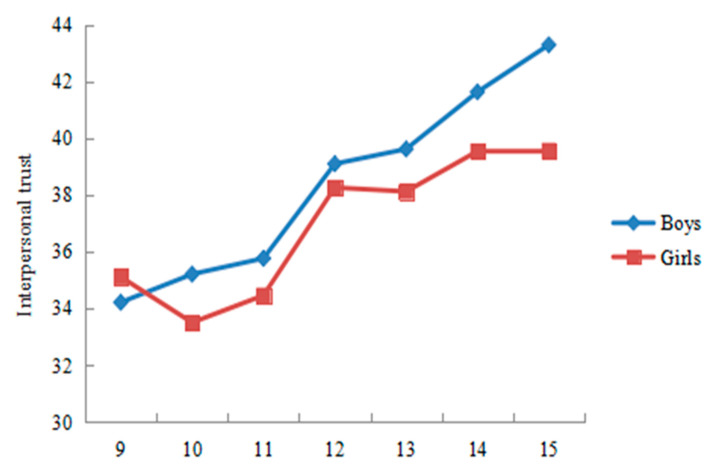
The total score of health-related quality of life by age and gender. Note: Higher score of interpersonal trust represents interpersonal mistrust.

**Figure 4 ijerph-19-15408-f004:**
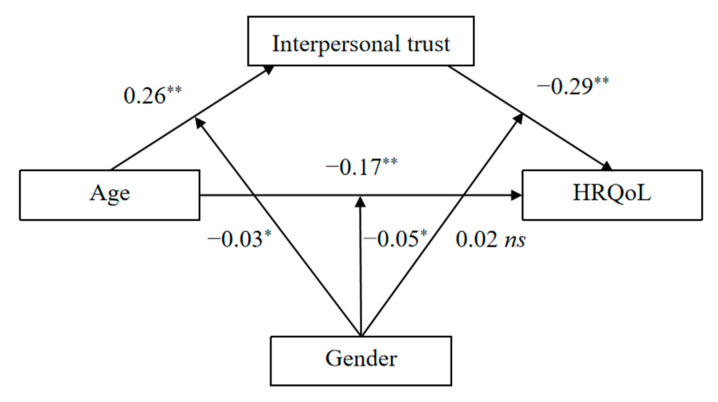
The observed moderated mediation model. Note: 1 = boys, 2 = girls; * *p* < 0.05, ** *p* < 0.01; Higher score of interpersonal trust represents interpersonal mistrust.

**Figure 5 ijerph-19-15408-f005:**
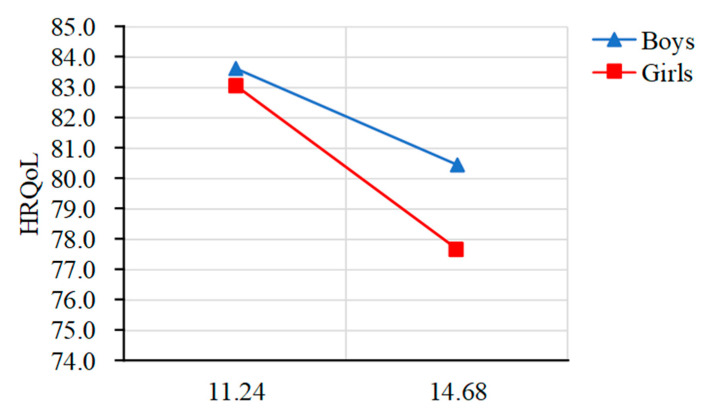
Gender moderated the relationship between age and health-related quality of life. Note: Higher score of interpersonal trust represents interpersonal mistrust.

**Figure 6 ijerph-19-15408-f006:**
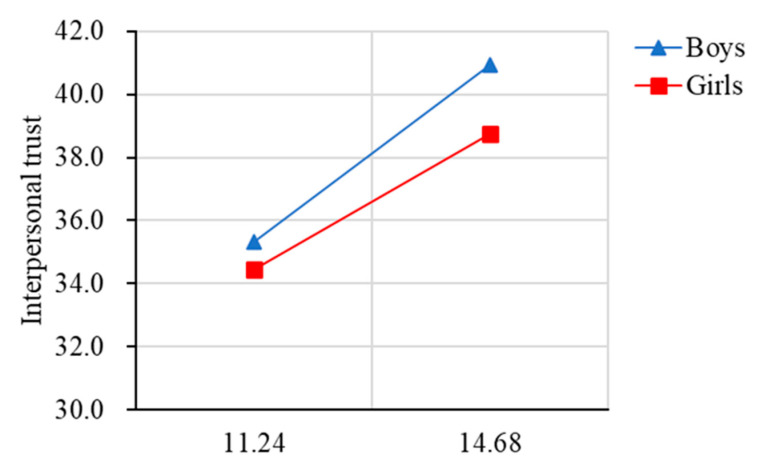
Gender moderated the relationship between age and interpersonal trust. Note: Higher score of interpersonal trust represents interpersonal mistrust.

**Table 1 ijerph-19-15408-t001:** Characteristics of participants (*n*= 6248).

	*n*	%
Age		
9	378	6.0
10	1036	16.6
11	1363	21.8
12	1106	17.7
13	996	15.9
14	852	13.6
15	517	8.3
Gender		
Male	2938	47.0
Female	3310	53.0

**Table 2 ijerph-19-15408-t002:** Means and standard deviations of study variables.

	Total		Boys		Girls			
	M	SD	M	SD	M	SD	*t*	*df*
Interpersonal trust	37.52	9.80	38.24	9.95	36.69	9.55	6.25 **	6215.54
HRQoL	81.24	12.32	81.75	12.17	80.70	12.45	3.24 **	6162.01
Physical function	82.46	14.12	83.49	13.88	81.32	14.29	6.00 **	6135.22
Emotional function	72.81	19.79	73.72	19.35	71.82	20.21	3.76 **	6110.26
Social function	88.46	13.92	88.04	14.12	88.92	13.69	−2.47 *	6246.00
School function	79.72	15.10	79.64	15.25	79.79	14.93	−0.38	6246.00

Note: * *p* < 0.05, ** *p* < 0.01; Higher score of interpersonal trust represents interpersonal mistrust.

**Table 3 ijerph-19-15408-t003:** Correlations between health-related quality of life and all variables.

	1	2	3	4	5	6	7	8
1 Age	1							
2 Gender	−0.02	1						
3 Interpersonal trust	0.25 **	−0.08 **	1					
4 HRQoL	−0.24 **	−0.04 **	−0.33 **	1				
5 Physical function	−0.14 **	−0.08 **	−0.24 **	0.85 **	1			
6 Emotional function	−0.22 **	−0.05 **	−0.27 **	0.81 **	0.54 **	1		
7 Social function	−0.14 **	0.03 *	−0.26 **	0.74 **	0.51 **	0.50 **	1	
8 School function	−0.27 **	0.01	−0.28 **	0.75 **	0.51 **	0.49 **	0.46 **	1

Note: * *p* < 0.05, ** *p* < 0.01; Higher score of interpersonal trust represents interpersonal mistrust.

**Table 4 ijerph-19-15408-t004:** Moderated mediating effect analyses.

Variables	Interpersonal Trust	95% CI	HRQoL	95% CI
Age	0.26 **	0.23, 0.28	−0.17 **	−0.20, −0.15
Gender	−0.08 **	−0.10, −0.05	−0.07 *	0.05, 0.09
Interpersonal trust			−0.29 **	−0.31, −0.26
Age × Gender	−0.03 *	−0.06, −0.01	−0.05 **	−0.07, −0.02
Interpersonal trust × Gender			0.02	−0.04, 0.06
*R* ^2^	0.07		0.14	
*F*	151.00 **		192.21 **	

Note: * *p* < 0.05, ** *p* < 0.01; Higher score of interpersonal trust represents interpersonal mistrust.

## Data Availability

The data presented in this research are available on request from the corresponding author.

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
