# Peer review of "Health-Related Quality of Life in Children: The Roles of Age, Gender and Interpersonal Trust"

_ijerph, 2022, doi:10.3390/ijerph192215408_

Round 1

Reviewer 1 Report

This manuscript uses mediation models and data on 6,248 Chinese students to examine the mediating and moderating effects of gender on associations between interpersonal trust and health-related quality of life (HRQoL). The data and models are appropriate to the task, and the question and results are interesting. I had a couple of suggestions.

--given how interesting the findings are, I felt like the paper could benefit from more detailed theoretical set up and pay off. As written, the manuscript essentially presents a problem and then gives the mathematical answer to that problem. There is so much more theoretical richness here that I feel not laying out those possibilities in the front half of the paper and then re-engaging with those possibilities in the discussion undercuts the paper's potential contributions. 

--there are relatively few control variables included in the models. While this allows for a clean test of moderating effects, it does bring up the question of whether any observed effects might be spurious. I would recommend that the authors address this question and justify their choices rather than trying to re-do models with more controls. 

--perhaps I am missing a note on this, but it is unclear whether the coefficients presented are standardized or unstandardized. I presume standardized, since that is typical for these kinds of models and the variables were standardized, but a brief mention of this as the results are presented would be a helpful clarification.

Author Response

Dear reviewer,

Thank you for giving us the opportunity to revise our manuscript. We appreciate for your helpful and constructive comments and suggestions, which are of great value to improving the quality of our article. Based on the comments, careful modifications have been made to our previous draft. In the current draft, all changes were highlighted within the document by blue colored text. After this revision, we also wrote a point-by-point cover letter to your comments.

We hope the revised manuscript will be acceptable for publication in the International Journal of Environmental Research and Public Health. Thank you very much for your time.

Reviewer 2 Report

Dear Authors,

I strongly suggest to research about teleology as a methodological error process. Not be aware of it drains a mayor part of the possible richness and efectivness of investigations. You were been working hard, but the research design also needs your valuable efforts. Also, your objectives and ethical issues must be clear and needs your attention.

Author Response

(The authors gave the same response as above.)

Reviewer 3 Report

First, I would like to congratulate the authors for taking up the topic of Health-related Quality of Life in Children: The Roles of Age, Gender and Interpersonal Trust in Children.

The work is very good in terms of technical and content. I have some notes that don't diminish the value of the work, but it would be worth editing some information:

1. The age-related description of the research group would be more interesting if, instead of a percentage description, it had a graph / diagram that would show the age of the children in percentage ranges.

2. I would ask for a clear, understandable separation of the purpose of the work and why such research is important for society. Descriptions and definitions are provided, but why it is so important, what the consequences are. Such short information about the purposefulness of such research should be included in the introduction to the work.

3. The methodology contained information that some children did not participate in the study and the group decreased from 7,000 to 6,248. Maybe it would be worth writing here why it happened, what were the exclusion criteria.

Author Response

(The authors gave the same response as above.)

Round 2

Reviewer 2 Report

Dear Authors,

You make important improvements about different concerns.

Now information is more clearly contextualized, there is a better proportion between research design and results.

I apreciate your effort. This new version can be an interesting propouse about chenese HrQoL  and pointed out about the importance of interpersonal-trust related to gender and age on global scale.